# Development and Laboratory Validation of Rapid, Bird-Side Molecular Diagnostic Assays for Avian Influenza Virus Including Panzootic H5Nx

**DOI:** 10.3390/microorganisms13051090

**Published:** 2025-05-08

**Authors:** Matthew Coopersmith, Remco Dijkman, Maggie L. Bartlett, Richard Currie, Sander Schuurman, Sjaak de Wit

**Affiliations:** 1Alveo Technologies, Inc., Alameda, CA 94501, USA; mattc@alveotechnologies.com; 2Department of R&D, Royal GD, 7418 EZ Deventer, The Netherlands; r.dijkman@gddiergezondheid.nl (R.D.); s.schuurman@gddiergezondheid.nl (S.S.); 3Johns Hopkins Bloomberg School of Public Health, Baltimore, MD 21205, USA; mbartl13@jh.edu; 4X-OvO: Poultry Diagnostics, Dunfermline KY118PB, UK; rcurrie@x-ovo.co.uk; 5Department of Population Health Sciences, Faculty of Veterinary Medicine, Utrecht University, 3584 CL Utrecht, The Netherlands

**Keywords:** avian influenza, bird-side, POC detection, poultry, H5, H7, H9, M gene, LAMP

## Abstract

Avian influenza A viruses (AIV) significantly impact both animal and human health. Reliable diagnostics are crucial for controlling AIV, including the highly pathogenic strains like H5Nx. In this study, we developed and validated the on-site Alveo Sense Poultry Avian Influenza Tests to rapidly detect the AIV M-gene and subtypes H5, H7, and H9 in unprocessed samples using reverse-transcription loop-mediated isothermal amplification (RT-LAMP) and impedance-based measurements. The Alveo Sense tests, using single-use microfluidic cartridges, deliver results within 45 min. Each cartridge includes assays for the AIV M gene and specific H5 and H7 or H9 subtypes, with internal process controls. The laboratory validation involved specificity, limit of detection (LoD), diagnostic sensitivity, reproducibility, and robustness tests using various AIV strains, other avian pathogens, and field samples. The assays showed 100% specificity for AIV subtypes without cross-reactivity with non-AIV pathogens. The LoD95 for H5, H7, and H9 ranged between RT-PCR Ct values of 29–33 in both cloacal and oropharyngeal samples and were able to detect avian influenza virus in both spiked samples and field samples. Reproducibility and repeatability studies showed perfect agreement across operators and laboratories and remained stable and accurate under different pre-analytical conditions. The Alveo Sense tests offer rapid, accurate, and reliable on-site diagnostics for AIV subtypes H5, H7, and H9 on samples from fresh dead and sick birds, valuable for early flock-level detection and outbreak control. Further field validation will improve the understanding of their diagnostic performance across various avian species.

## 1. Introduction

Influenza A viruses are increasingly important pathogens involving animal and human health. Aquatic birds are the primary reservoir for avian influenza viruses (AIV), which are commonly classified into subtypes based on the surface glycoproteins haemagglutinin (H) and neuraminidase (N). To date, sixteen HA subtypes and nine NA subtypes have been identified in birds [1]. In poultry, the AIV strains are classified into two pathotypes: low pathogenicity (LP) and high pathogenicity (HP). The number of basic amino acids in the HA0 cleavage site plays a critical role in pathogenicity, determining which proteases can cleave HA and in which tissues the AIV can replicate [2]. LPAIV usually have a cleavage site that contains less than two basic amino acids in a critical position and are therefore cleaved by trypsin or trypsin-like proteases, limiting replication of these viruses principally to epithelial cells of the intestinal and respiratory tracts of birds. By contrast, a multi-basic cleavage site contains several basic amino acids at the same critical position and is cleaved by several common cellular proteases present in most cells throughout the body, causing systemic disease and lethal infection of high-pathogenicity AIV (HPAIV) in gallinaceous poultry species [2].

Until early 2000, all AIV strains detected in aquatic birds were of low pathogenicity resulting in minimal to no clinical signs during infection [3]. All naturally occurring H1-4, H6, and H8-16 AI viruses have been of low pathogenicity for chickens when challenged by the respiratory route [4]. As the name indicates, the virus causes little mortality, but the damage caused by respiratory infections and drops in egg production might still be very significant [5,6,7]. H9N2 is an example of a LPAI virus that causes major damage to chickens in large parts of Asia, the Middle East and North Africa.

H5 and H7 avian influenza strains can also be divided into low and high pathogenicity, these latter being the cause of Bird Flu, a deadly disease in poultry. Historically, HPAI H5 or H7 strains resulted from mutations in the haemagglutinin cleavage site of LPAI ancestor strains during an infection in gallinaceous poultry such as chicken, turkey, pheasant, and quail. In these cases, culling of the HPAI infected birds effectively led to the eradication of the HPAI virus as they did not get established in wild birds. This remains true for the H7 subtype. However, for subtype H5, the situation has changed dramatically in recent decades since the appearance of the goose Guangdong (Gs/GD) H5Nx lineage in China in 1996 [3]. The original A/goose/Guangdong/1/1996 H5N1 virus has evolved into a highly successful lineage of H5Nx viruses which began spreading in waves around the world via migrating wild birds starting in 2003. After spreading across parts of Asia, the Middle East, Europe, and North America, the first cases were identified in Central and South America in 2022, followed by Antarctica in 2024. This pandemic of Gs/GD H5Nx viruses in wild birds has led to mass deaths in an increasing number of bird species, major outbreaks of bird flu in poultry, and deaths in marine and various land mammals and poses an increasing risk of human infection. Most recent is the large-scale spread of H5N1 in the dairy industry in the United States of America.

Reliable diagnostic tools that are fit for purpose are essential for the control of AIV of any subtype, both in unvaccinated and in vaccinated flocks [8]. The diagnostic tools used for the detection of AIV or to show the freedom of AIV are predominantly performed in a laboratory setting and can be divided into techniques that detect the virus or the antibody response. Virus detection by RT-PCR or antigen detection by ELISAs are performed under strict laboratory conditions that offer well controlled, standardized, and clean conditions and the dedication of well-trained laboratory staff [9]. Another advantage of laboratory testing is that sample preparation can be optimized for the type of sample when needed. In AIV testing, the common types of samples are oropharyngeal (OP) and cloacal swabs (CL), or organ tissue. In the field, rapid “Point-of-Care” (POC) tests are also used for the detection of AIV, which can provide almost immediate results but are performed under conditions that are out of control.

In general, RT-PCR is considered to have a more or less comparable detection limit to the gold standard virus isolation [10,11,12,13], whereas antigen capture immunoassays vary in sensitivity and usually have a detection limit that is at least 3–4 log_10_ EID_50_ higher than virus isolation [14,15]. These antigen capture assays are AIV specific, but they provide no information of the subtype of AIV involved.

The selection of which test to use depends on a number of factors that include the purpose of testing and the sample kind and origin. The sample’s viral load is also variable and is highly dependent on the phase of the infection, involved virus strain, animal species, sample type, and sampled organ or tissue [16]. While testing of fresh dead or a sick bird that is in the acute phase of infection does not require a very low detection limit as a high amount of virus is expected, testing to show an absence of virus in healthy birds requires high sensitivity. The availability of a reliable, highly specific AIV subtype-specific on-site test with a detection limit lower than the present on-site lateral flow tests would improve the quality of the diagnosis in cases of AIV suspicion.

Here, we report the development and results of the laboratory validation of the on-site Alveo Sense Poultry Avian Influenza Tests for the detection of AIV M-gene and genomes of subtype H5, H7, and H9 viruses in unprocessed cloacal and oropharyngeal samples.

## 2. Materials and Methods

### 2.1. The Alveo Sense Technology

The Alveo Sense Poultry Avian Influenza Test Type A H5 H7 (AIV/H5/H7) and the Alveo Sense Poultry Avian Influenza Test Type A H5 H9 (AIV/H5/H9) are multiplexed rapid molecular tests that utilize reverse-transcription loop-mediated isothermal nucleic acid amplification (RT-LAMP) technology and electrical impedance sensors to qualitatively detect and differentiate respective avian influenza viral RNA strains. Combining RT-LAMP with impedance-based measurements eliminates the need for thermal cycling, fluorescent probes, and optics, enabling portability. These tests are performed using a single-use microfluidic cartridge, delivering results in approximately 45 min or less (Figure 1).

Each cartridge is equipped with six assays designed for broad detection of the Avian Influenza A virus through the M gene, as well as precise detection of H5 and H7 or H9 subtypes, depending on the cartridge type. The cartridge also contains an internal process control to ensure the validity of each result. The Alveo Sense Poultry Avian Influenza Tests provide qualitative results from samples collected from either the oropharynx or cloaca and have been validated to accommodate pooled samples of up to five samples for cloacal testing and up to ten samples for oropharyngeal testing.

To begin testing, the user elutes up to five cloacal swabs or up to ten oropharyngeal swabs into the Swab Elution Vial. The sample is then reverse filtered to remove particulates. Then, 500 µL of the filtered sample is added to the Assay Buffer vial and mixed by inversion. The sample is transferred into the microfluidic cartridge, resuspending the dried reagent and preparing the test for use. The Alveo Sense Poultry AIV RT-LAMP test is performed at 52 °C for 10 min (reverse transcription), followed by 65 °C for 35 min (isothermal amplification) (Figure 2). Testing is controlled via the Alveo Sense Mobile Application, which connects to the analyzer via Bluetooth (Figure 1).

### 2.2. LAMP Assay Development

For the H5, H7, and H9 assay design, Influenza A virus HA sequences were obtained from the GISAID EpiFlu™ database, matching the following criteria: Influenza A virus of avian origin; serotype H5, H7, or H9, respectively; collected from 2018 to 2023; and globally sourced. Only sequences from original samples with a complete segment 4 sequence (HA) were included.

For the generic M-targeted Influenza A assay, sequences were also obtained from the GISAID EpiFlu™ database with the following criteria: Influenza A virus of avian origin; collected from 2018 to 2023; and globally sourced. Only sequences with a complete segment 7 (M) were included.

Multiple sequence alignments were performed using ClustalO (version 1.2.2) to identify conserved regions specific to each subtype for HA or shared among all strains for M. RT-LAMP primers were designed either manually or using the NEB LAMP Primer Design Tool (version 1.4.1). Each primer design, except for H9, consists of seven specific primers: two inner primers (FIP and BIP), two outer primers (F3 and B3), two loop primers (LF and LB), and a reverse transcription primer (RT). The H9 primer set includes all aforementioned primers except the backwards loop primer (LB). Primer secondary structures and potential dimers were checked either using the OligoAnalyzer Tool (IDT, Coralville, IA, USA) version 3.1. April 2023 or Oligo Calc: Oligonucleotide Properties Calculator (version 3.10).

The specificity of the designed primers was checked in silico by mapping them to the downloaded H5, H7, H9, and M sequences using the following criteria: a maximum of three mismatches allowed in the binding region, with no mismatches within 3 bp of the 3′ end. Primer sequences were manually adapted as needed for optimal subtype, lineage, or clade coverage and specificity. Additionally, a primer BLAST (based on Primer3 version 2.5.0 against the NCBI nucleotide database was performed to ensure there was no cross-reactivity with non-target sequences possible.

### 2.3. Strains and Sample Collection and Matrix Preparation

A diverse panel of archival strains was used to validate the test protocols, encompassing a variety of avian influenza virus (AIV) strains, infectious bronchitis virus (IBV) strains, and other avian pathogens (both viruses and bacteria). The specific strains and pathogens included are shown in Table 1 and Table 2 and are mentioned under laboratory validation. The archival samples were stored at −80 °C and thawed prior to testing. Samples were processed according to standard laboratory protocols to ensure consistency and reliability. Each sample underwent real-time PCR testing to determine correct identification and estimation of viral or bacterial load. The PCR assays used targeted specific genes relevant to each pathogen, ensuring accurate detection and quantitation. Assays were conducted using established protocols, with controls included to validate the results.

Swab samples used for test validation were collected by Royal GD (Deventer, Netherlands) from specified-pathogen free (SPF) chickens sacrificed in earlier experiments. To collect the oropharyngeal and cloacal samples, swabs were inserted into the appropriate orifice and rotated five times to ensure sufficient mucosal material was obtained. The swabs were collected from July 2024 to September 2024 and were stored frozen at temperatures below −70 °C until ready for sample elution. The collected matrix was used during testing to simulate the inhibitors found in real-world field samples.

For Limit of Blank (LoB), Limit of Detection at 95% (LoD95), Reagent Stability, and Reproducibility and Repeatability studies, the SPF swabs were eluted into the swab elution solution at a pooling ratio of two swabs per 1 mL for oropharyngeal samples and one swab per 1 mL for cloacal samples. The swab elution vial provided with the Alveo Sense AIV/H5/H7 and AIV/H5/H9 kits contain 5 mL of the swab elution solution, and thus the pooling ratio conducted in these studies reflect 10 swabs per vial for oropharyngeal samples and 5 swabs per vial for cloacal samples.

To accommodate Limit of Blank testing requirements, four unique pools of cloacal and oropharyngeal matrices were produced. To minimize health and safety risks, the clinical pools were inactivated in a water bath for a minimum of 15 min at 60 °C and a minimum of 25 min at 95 °C. To minimize confounding variables, the bulk matrices used for these studies were filtered in their entirety before testing. Aliquots of the individual pools were prepared and frozen to cover the LoB testing demands. The remaining volumes of the unique cloacal and oropharyngeal pools were combined to create one large pool for the remaining studies. Cloacal and oropharyngeal matrices were prepared and tested separately.

For the technical specificity studies, the cloacal and oropharyngeal pools were prepared by eluting either five cloacal swabs per swab elution vial or ten oropharyngeal swabs per swab elution vial. Each matrix suspension was then reverse-filtered using the syringe filter provided with the test kit, and a total of 3.5 mL of the filtered matrix was added to 31.5 mL of the assay buffer to maintain the 10-fold matrix dilution.

### 2.4. Influenza a Real-Time PCR

RNA was extracted from the different samples used in this study with the MagMax RNA/DNA isolation kit on a KingFisher™ Flex Purification system (Thermo Fisher Scientific, Waltham, MA, USA). A generic PCR, as described by Ward et al. [17] with some modifications and targeting the Influenza A matrix gene, was used for the detection of Influenza A virus. In brief, the extracted RNA was tested using the AgPath-ID™ One-Step RT-PCR kit (Thermo Fisher Scientific) on a QuantStudio™ 5 system (Thermo Fisher Scientific) with the following program: 10 min at 45 °C, 10 min at 95 °C, followed by 45 cycles of 15 s at 95 °C, and 45 s at 60 °C.

### 2.5. Laboratory Validation

Limit of Blank (LoB) testing was performed in accordance with CLSI EP17-A2, where LoB is defined as the “highest measurement result that is likely to be observed (with a stated probability, α) for a blank sample.” This study involved testing four unique cloacal pools and four unique oropharyngeal pools, with each pool tested in five replicates, resulting in 20 valid results per matrix type and per Alveo Sense assay (AIV/H5/H7 or AIV/H5/H9).

Technical specificity of the assays was evaluated across different Influenza A viruses and other non-influenza pathogens to assess the detection of various Avian Influenza strains and to verify no cross-reactivity with non-avian influenza microbial organisms. The specificity testing included the following twelve Influenza A strains: H3N1, H5N1 (European non-GsGD), two H5N2 strains (one EU-non-GsGD, one AM-non-GsGD), two H5N3 strains (both EU-non-GsGD), H5N8 (2.3.4.4b), H6N1, H7N1 (Eurasian lineage), H7N7 (Eurasian lineage), and two H9N2 strains (G lineage). These influenza strains were tested in both cloacal and oropharyngeal matrices, with a minimum of three cartridge replicates. Ct values of the influenza strains used ranged from Ct24–32 (Table 1). Additionally, DNA and RNA from the following nineteen non-influenza viruses and bacteria were included in the specificity testing: four strains of Infectious bronchitis virus (D388_GI-19, D274_GI-12, M41_GI-1, and 4/91_GI-13), Infectious laryngotracheitis virus, avian metapneumovirus subtypes A, B, and C (AMPV-A, AMPV-B, AMPV-C), Newcastle Disease virus (NDV), two *Mycoplasma synoviae* strains, two *Mycoplasma gallisepticum* strains, three *Avibacterium paragallinarum* serovars (A1, B1, and C4), *Pasteurella multocida*, *Gallibacterium anatis*, *Ornithobacterium rhinotracheale*, and *Riemerella anatipestifer*. Testing was performed in duplicate in the presence of the oropharyngeal matrix. To ensure no cross-reactivity would occur at high viral and bacterial loads, positive samples were selected with Ct values ranging from Ct14–20, as determined in their specific PCRs. Testing of these pathogens with the AIV/H5/H7 and AIV/H5/H9 cartridges targeted a Ct range of approximately 20–25 in the final reaction. For two bacterial samples (*Ornithobacterium rhinotracheale* and *Riemerella anatipestifer*), a > 4.0 McFarland suspension was used without determination of the Ct value.

Limit of Detection (LoD95) was defined as the lowest concentration where ≥95% of the replicates tested positive for the HA-specific test (H5, H7, H9). This experiment was conducted using four viral samples: an H5N2 strain of the American non-GsGD clade, an H5N8 strain of the 2.3.4.4b clade, an H7N7 strain of the Eurasian lineage, and an H9N2 strain of the G lineage. LoD testing was conducted in two phases: Preliminary LoD and Confirmatory LoD. Preliminary LoD involved range-finding with a minimum of three concentrations tested with five replicates each. Confirmatory LoD involved testing fifteen additional replicates at the lowest concentration that yielded 100% detection during the Preliminary phase. The LoD was confirmed if the H5, H7, or H9 indicator achieved a ≥95% positivity rate across 20 valid cartridge replicates (≥19/20). Each sample (H5N2, H5N8, H7N7, H9N2) was tested in the presence of both cloacal and oropharyngeal matrices, and results were compared with those obtained by TaqMan real-time PCR following the description under Influenza A real-time PCR.

A three-day reproducibility and repeatability experiment was conducted to evaluate precision and test consistency. Testing was performed across two laboratories using twelve Alveo analyzers divided among four operators over three days. Each operator tested three replicates per condition per day, including a no template control (NTC) and a combined positive control (PC). For the AIV/H5/H7 assay, the PC included H5N2, H5N8, and H7N7 strains, while the PC for the AIV/H5/H9 assay included H5N2, H5N8, and H9N2 strains, each at a concentration of 3x LoD. All testing was conducted in the presence of cloacal matrix. This study design resulted in a total of 36 negative and 36 positive replicates per cartridge type (AIV/H5/H7 and AIV/H5/H9).

The pre-rehydration shelf life of an AIV cartridge with dried-down reagents is currently established at 12 months at a storage condition of 2–30 °C (similar to the ALVEO be well COVID-19 assay) with ongoing stability studies in progress. Post-rehydration reagent stability of the Alveo Sense AIV/H5/H7 and AIV/H5/H9 assays was determined by filling the cartridge with Assay Buffer and sample (either a combined H5N2/H5N8/H7N7, a combined H5N2/H5N8/H9N2, or a negative sample) and leaving them for 0, 10, 15, 20, or 25 min at 30 °C before inserting the cartridge into the analyzer and starting the test. This study was intended to verify that the assays would remain stable and provide accurate results under various pre-analytical conditions.

### 2.6. Positive Field Sample, Positive Spiked Sample, and Negative Field Sample Testing

Known positive and negative field samples, along with spiked positive samples, were tested to validate the system, workflow, and test performance. Oropharyngeal swabs from eight influenza-negative flocks, collected as part of early warning monitoring, were pooled at a ratio of five swabs per vial per flock and tested with a minimum of two replicates following the end-to-end workflow. All field samples were confirmed negative by the Wageningen BioVeterinary Research reference institute laboratory prior to testing. Additionally, oropharyngeal and cloacal swabs from an H5N1 LPAI (EU_nonGsGD) positive flock were collected and tested using the complete workflow.

Due to avian influenza testing regulations, a true clinical validation using multiple positive samples was not feasible in the laboratory setting. Therefore, positive sample validation was performed by spiking the viruses used in the LoD experiments (H5N2, H5N8, H7N7, H9N2) into the sample matrix prior to testing. Positive sample testing was concurrently evaluated with pool size validation: oropharyngeal samples were tested in pools of 1, 3, 5, and 10, while cloacal samples were tested in pools of 1, 3, and 5. Each unique condition was tested with a minimum of two cartridge replicates.

### 2.7. Statistical Analysis

Cohen’s kappa was used to calculate agreement between expected and observed results.

### 2.8. Ethical Statement

During the development and evaluation of the tests, no human samples were used. No specific permission from the animal welfare body was required as the OP and CL swabs used for LoB, LoD95 and specificity testing had been collected from euthanized birds that had been sacrificed in earlier experiments. The OP and CL swabs from the LP H5N1 infected flock had been taken for diagnostic purposes.

## 3. Results

### 3.1. LAMP Primer Design

For the development of the M assay, a total of 1474 full M sequences were analyzed. These sequences represented a broad range of serotypes, including H3N8, H5N1, H5N2, H6N6, H7N3, H7N7, and H9N2. Twelve M designs were initially selected for testing and optimization. Based on technical performance and broad detection capabilities, two assays were ultimately selected to maximize specificity. These two assays target different regions of the M gene and complement each other in coverage, thereby reducing the risk of false negative results due to primer mismatches.

For the development of the H5 assay, 9532 full HA sequences were included in the analysis. Sequence alignment was performed, and a maximum-likelihood phylogenetic tree was constructed, revealing clear clusters aligned with the currently defined H5 clades. According to the H5 clade definition, the sequences were classified as follows: 2.3.2.1a (n = 177), 2.3.2.1c (n = 150), 2.3.4.4b (n = 8356), 2.3.4.4c (n = 66), 2.3.4.4e (n = 15), 2.3.4.4g (n = 68), 2.3.4.4h (n = 463), Am_nonGsGD (n = 147), and EA_nonGsGD (n = 90). It became evident during development and optimization that a single H5 assay would not sufficiently cover the genetic diversity, posing a risk for false negative results. Out of 48 different H5 LAMP designs tested for performance and specificity, 3 designs—H5A2e, H5_set19, and H5G—were ultimately selected in order to provide comprehensive coverage of the relevant clades and lineages. The H5A2e set broadly detects different H5 clades, except for the AM_nonGsGD. H5_set19 is optimized for the globally prevalent clade 2.3.4.4b. H5G is tailored for the AM_nonGsGD clade.

For the H7 assay development, 384 sequences were included. The maximum-likelihood phylogenetic tree showed clear distinctions among North American (n = 152), South American (n = 19), Oceanian (n = 5), and Eurasian (n = 205) lineages. The Eurasian lineage was further subdivided into two sublineages: Eurasian-African (n = 92) and H7N9 Asian (n = 114). Out of 39 designs tested, one H7 design was selected for its broad detection of H7 strains.

For the H9 assay development, 3856 sequences from GISAID were included. The maximum-likelihood phylogenetic tree indicated that these sequences belonged to the following H9 lineages: lineage B (clades B3, B4, B4.5, B4.6, and B4.7; n = 2901), lineage Y (clades Y2, Y3, Y4, Y6, Y8, Y9, and Y2.2; n = 133), and lineage G (clades G4, G5, G5.3, G5.5, G5.6, G5.7, and G5.3.2; n = 822). Out of nine H9 designs selected for further development and optimization, one set was chosen for inclusion in the final cartridge. The selected H9 design demonstrated the broadest detection and specificity across the various H9 lineages.

### 3.2. Limit of Blank (LoB)

Limit of Blank testing was performed on four unique oropharyngeal and cloacal matrix pools, with each pool tested in five replicates per cartridge type (AIV/H5/H7 and AIV/H5/H9). The two M targets, three H5 targets, and the H7 and H9 targets all showed 100% negativity rate across the influenza-negative samples (40 correctly identified negative results where 40 were expected for AIV/H5/H7 and 40 correctly identified negative results where 40 were expected for AIV/H5/H9). This demonstrates that there is no cross-reactivity with other RNA or DNA (both prokaryotic and eukaryotic) that may have been present in these samples and suggests a low risk of Type 1 error for true-negative results.

### 3.3. Technical Specificity

For both cloacal and oropharyngeal samples, we found 100% specificity using both the AIV/H5/H7 and the AIV/H5/H9 tests. The six Influenza H5 strains were tested with a minimum of three replicates per sample type and were specifically detected by the M and H5 assays, showing no reactivity from H7 or H9. The two Influenza H7 strains were exclusively detected by the M and H7 assays, showing no reactivity from H5 or H9. The two Influenza H9 strains were detected solely by the M and H9 assays, showing no reactivity from H5 or H7. The Influenza H3N1 and H6N1 strains were detected only by the M assay and were negative for H5, H7, and H9 (Table 1). None of the respiratory non-influenza viruses and bacteria (tested in duplicate per AIV/H5/H7 and AIV/H5/H9) were detected across any M, H5, H7, or H9 assay (Table 2).

Overall, this resulted in 135 correctly identified influenza-positive results, where 135 were expected, and 76 correctly identified negative samples, where 76 were expected. With a Cohen’s kappa value of 1.0, indicating perfect agreement between expected and observed results, this demonstrates very high specificity of the assays and shows there is no cross-reactivity with the non-influenza viruses and bacteria tested.

### 3.4. Limit of Detection (LoD95)

Strain H5N2 (clade Am_nonGsGD) was consistently detected in 100% (20/20) of cloacal sample replicates down to a real-time PCR equivalent of Ct33.5 for both M and H5 targets. Detection remained high at Ct35.1, with 100% (5/5) for M and 60% (3/5) for H5. In oropharyngeal samples, it was detected in 100% (20/20) of replicates at Ct33 for both targets, dropping to 100% (5/5) for M and 40% (2/5) for H5 at Ct34.5. These results indicate an LoD95 of Ct33 of the assay for this strain in both sample types.

Strain H5N8 (clade 2.3.4.4b) was found in 100% (20/20) of cloacal sample replicates down to Ct30.6 for both M and H5 targets, maintaining 100% (5/5) detection for M and 80% (4/5) for H5 at Ct32.3. In oropharyngeal samples, detection was 100% (20/20) at Ct31.9 for both targets, decreasing to 35% (7/20) for M and 85% (17/20) for H5 at Ct33.6, and detected in 1 of 5 replicates for H5 at Ct35.2. The LoD95 of the assay for this strain ranges between Ct31–32.

Detection of strain H7N7 (Eurasian lineage) was 100% (20/20) in cloacal samples down to Ct28.5 for both M and H7 targets. At Ct30.0, 80% (4/5) detection was observed for both M and H7 targets, and at Ct 31.5, the assays observed a detection rate of 80% (4/5) for M and 40% (2/5) for H7. In oropharyngeal samples, 100% (20/20) detection occurred at Ct29.0 for both targets, reducing to 100% (5/5) for M and 60% (3/5) for H7 at Ct30.5, and 80% (4/5) for M and 60% (3/5) for H7 at Ct32.1. The LoD95 of the assay for this strain ranges between Ct29–30.

Strain H9N2 (G lineage) showed 95% (19/20) detection in cloacal samples for M and 100% (20/20) for H9 targets down to Ct31.6. At Ct33.2, detection was 60% (3/5) for both targets. In oropharyngeal samples, detection was 85% (17/20) for M and 100% (20/20) for H9 at Ct31.6, dropping to 1 of 5 replicates for M and 60% (3/5) for H9 at Ct33.2. The LoD95 of the assay for this strain is between Ct31–32.

### 3.5. Precision Analysis

We assessed the reproducibility and repeatability of our testing procedure across different operators and laboratories. For each cartridge type (AIV/H5/H7 and AIV/H5/H9), three positive and three negative samples were tested over three different days by two operators in each of two labs. The results were very consistent, yielding 100% agreement with the expected result across all test concentrations, operators, laboratories, days, and cartridge types (72 correctly identified influenza-positive results where 72 were expected, and 72 correctly identified negative samples where 72 were expected). The Cohen’s kappa value of 1.0 further confirms perfect agreement, demonstrating the robustness and reliability of our testing procedure.

### 3.6. Robustness

Post-rehydration reagent stability testing was conducted for the AIV/H5/H7 and AIV/H5/H9 assays. The results showed that negative samples remained negative even after the reaction mix was fully reconstituted and subjected to prolonged incubation at 30 °C for intervals of 10, 15, 20, and 25 min before initiating testing (30 correctly identified influenza-negative results where 30 were expected). Additionally, combined positive samples for both the AIV/H5/H7 and AIV/H5/H9 tests were reliably detected throughout the duration of the experiment in their respective assays (30 correctly identified influenza-positive results where 30 were expected). This demonstrates that although it is advised to start the reaction immediately after loading the cartridge, the test’s robustness allows it to accommodate variability that may arise in a field setting. While our findings indicate strong robustness, the process control embedded in each cartridge would yield an invalid result in the event of major operator errors or sample-related issues.

### 3.7. Positive Field Sample, Positive Spiked Sample, and Negative Field Sample Testing

The confirmed influenza-negative oropharyngeal swabs from eight influenza-negative poultry flocks all tested negative for M, H5, H7, and H9 in the AIV/H5/H7 and AIV/H5/H9 assays, while the internal control was detected in all tests, ruling out sample-related inhibition and confirming the true negative results.

The results of the AIV spiked samples for the AIV/H5/H7 assay are presented in Table 3, showing robust detection in the majority of samples. For the samples containing the H5N2 (AM_nonGsGD), the Ct values ranged from 29 to 36. All samples were detected in both the M and H5 assays, except for one oropharyngeal sample (pool of 10) with a Ct value of 36, which was detected by M in 2 out of 2 replicates and by H5 in 1 out of 2 replicates. The samples containing H5N8 (2.3.4.4b) exhibited Ct values ranging from 28 to 31. All samples tested positive for both M and H5, with the exception of one cloacal sample (pool of 5) with a Ct value of 30, which was detected by H5 in 3 out of 3 replicates and by M in 2 out of 3 replicates. All H5N2 and H5N8 tests were negative for H7. For the samples with H7N7 (Eurasian lineage), Ct values ranged from 24 to 30. All oropharyngeal and cloacal samples tested positive for M and H7 and were negative for H5.

The results of the AIV spiked samples for the AIV/H5/H9 assay are presented in Table 4, showing consistent detection across all samples. The samples containing H5N2 (AM_nonGsGD) demonstrated Ct values ranging from 30 to 33. All samples tested positive for both M and H5, and negative for H9 as expected. The samples with H5N8 (2.3.4.4b) exhibited Ct values ranging from 28 to 30, with all samples testing positive for M and H5, and negative for H9. Samples containing H9N2 (G lineage) showed Ct values ranging from 26 to 29. All oropharyngeal and cloacal samples tested positive for M and H9 and remained negative for H5.

Oropharyngeal and cloacal samples from animals infected with low pathogenic avian influenza (LPAI) H5N1 were tested using the Alveo Sense AIV/H5/H7 assay. All three cloacal samples, with Ct values of 23, 25, and 30 in the real-time PCR, tested positive for both M and H5 using the Alveo Sense assay. Among the five oropharyngeal samples tested, which had Ct values of 20, 27, 29, 30, and 31 in real-time PCR, three samples (Ct 20, 27, and 31) tested positive for both M and H5. The remaining two samples (Ct 29 and 30) tested positive for influenza A (M positive) but were negative for H5.

## 4. Discussion

The use of reliable, validated diagnostics that are suitable for the purpose for which they serve is one of the key requirements in the fight against avian influenza. In cases where clinical signs indicate possible AIV infection, tests that detect the virus are more appropriate than serological tests. In addition to test quality, the availability and speed of obtaining the results are also important factors to consider. Despite the advantages of RT-PCR for the detection of the AIV genome, several drawbacks are associated with its use. RT-PCR must be performed by well-trained technicians using specialized equipment under controlled lab conditions, making it relatively costly and less accessible in resource-limited settings or remote areas where suitable laboratories are not available. Furthermore, longer transport times can delay results. Non-PCR-based Nucleic Acid Amplification Technologies (non-PCR-NAAT) such as LAMP require less elaborate sample preparation, eliminate the need for thermal cycling, and offer shorter turnaround times [9]. A rapid and subtype-specific diagnosis of an AIV infection helps take the right measures to quickly contain a potential or active outbreak.

The technology and assay detailed in this paper have overcome several of the well-known limitations relating to LAMP assay development and its practical application in the field setting. LAMP-based assays have historically been prone to false positives due to aerosol contamination and the generation of nucleic acid products. However, the developed cartridge structure creates a closed system that contains the amplicon produced during the reaction, significantly reducing the risk of contamination. Furthermore, while LAMP assays often face challenges of non-specific amplification with multiplexed targets due to the large number of required primers, the cartridge’s ability to test multiple individual targets simultaneously minimizes this risk while providing clinically useful information to the end user. The advanced microfluidics of the cartridge also ensures uniform distribution of reaction chemistry across the multiple assay targets. Furthermore, the development of a portable, robust reader with self-contained heating capabilities eliminates the need for any additional equipment on-site, and the analyzer’s aluminum housing allows for easy cleaning after each use. Finally, the mobile application’s step-by-step instructions make testing easy for new and experienced users, and the use of impedance measurements with complex algorithm analysis as a detection system removes any subjectivity in determining the assay endpoints and overall qualitative result. These details combine to create the user experience of an assay that produces accurate, repeatable results with minimal operator training. In addition to the pen-side test, the Alveo Sense platform includes a secure results database—Alveo Vista—that provides real-time access to up-to-date information through time-stamped and geo-tagged results. All data are stored in an encrypted portal, which can support varying levels of access to organizations and public health officials. This combination of early testing and centralized data monitoring can complement existing global surveillance networks and enhance monitoring programs.

During development of the M, H5, H7, and H9 assays, it became evident that two or three complementary assays were needed to cover the extensive genetic diversity of the M and H5 genes. As LAMP uses 4–7 primers for the specific amplification of target DNA, primer design is more complex than for the RT-PCR. This complexity requires careful consideration to ensure that each primer and assay work effectively. The necessity of using multiple primer sets to detect all desired strains has been previously reported with RT-PCR by Laconi et al. [11]. They compared the sensitivity and specificity of four published protocols using a single set of primers and probes for the detection of the M gene, and a fifth commercial kit that used a combination of primers and probes targeting both the M and NP genes. Only the RT-PCR kit that used the combination of M and NP primer/probes was able to detect all 152 AIV strains, with the sensitivity of the other four RT-PCR assays ranging from 89.9% to 98.7%. These findings show that consistent monitoring of the reported sequences of the M and H genes is needed to ensure that the selected primers and probes used in genomic tests as RT-PCR and LAMP assays are capable of detecting all relevant strains.

The evaluation of the Alveo Sense Poultry Avian Influenza H5, H7, and H9 subtype tests showed a technical specificity of 100% in combination with high repeatability and robustness for both OP and CL samples. Comparing the detection limit of the Alveo Sense Poultry Avian Influenza tests and RT-PCR for the M gene showed that the Alveo Sense tests had a limit of detection corresponding to a Ct range of approximately 30–32, with 100% agreement with RT-PCR at lower Ct values. However, as the Ct values increased, the sensitivity of the Alveo Sense tests decreased rapidly. Filaire et al. [18] reported a comparable LoD of a Ct of around 30 resembling 5.75 to 9.65 viral copies per μL of two 2.3.4.4b HP H5 viruses for a RT-LAMP assay specifically developed for the detection of 2.3.4.4b HPAI H5 strains. A similar trend was observed when comparing the detection limits of two lateral flow tests (LFDs) with RT-PCR: both LFDs were positive when the Ct was below 23 in OP and CL samples but became negative as the Ct values increased [19]. Comparable results were reported by Soliman [20], who tested two LFDs and found a specificity of 90% for both LFDs, with an analytical sensitivity limit of approximately 10^5^ EID50/mL.

For the field, diagnostic sensitivity is more important than the detection limit. For swabs taken from fresh dead or sick animals with a high viral load (e.g., average Ct value of 20), many tests will demonstrate sufficient to high diagnostic sensitivity. However, if these samples contain a low viral load (e.g., Ct values of 35–40), only RT-PCR will be able to provide good diagnostic sensitivity. Correct sampling is thus very important. Publications reporting the amounts of AI virus found in oropharyngeal, tracheal, and cloacal swabs of unprotected, affected animals usually show a range between 10^4.5^ EID_50_ and 10^8.5^ EID_50_/mL for HP H5 and H7 [21,22,23,24,25,26,27,28,29,30], and between 10^2^ EID_50_ and 10^6^ EID_50_/mL for subtype H9 in OP swabs [31,32]. Nguyen et al. [33] found 10^7^ to 10^8^ H9N2 RNA copies/mL in the respiratory tract and 10^4^ to 10^6^ H9N2 RNA copies/mL in the cloaca. Studies at Royal GD with two G1 H9N2 strains showed M RT-PCR Ct values between 24 and 30 in the respiratory tract (De Wit, unpublished studies).

Based on the RT-PCR results of the titration series used in the LoD determination of the Alveo Sense assay, we expect the tipping range for the test to fall between 10^2^ and 10^3^ EID^50^/^mL^ for both OP and CL swabs. Given these results, the diagnostic sensitivity of the Alveo Sense AIV/H5/H7 and AIV/H5/H9 assays is expected to be high in samples derived from fresh dead or sick birds. Furthermore, the Alveo Sense AIV/H5/H7 and AIV/H5/H9 tests achieved a combined validity rate of 99.8% (938 valid results out of an expected 940 valid results), demonstrating the tests’ reliability. While extensive laboratory testing has been performed on the Alveo Sense tests, the real diagnostic sensitivity and specificity can only be determined under field conditions by testing samples from healthy and sick or dead animals. The first results of the Alveo Sense test on the field samples collected in the acute phase of the LP H5N1 outbreak give confidence, but the dataset needs further expansion. When expanding the quantity of field data, it is also of importance to include samples from other animal species such as turkeys, ducks, geese, and mammals in order to know what the test can be used for with high reliability. Depending on the results and the requirements for diagnosing AIV infection, it can then be decided where the test is best used, either as a screening test or as a decisive test. In the event of various questions and circumstances, the further use of confirmatory tests such as RT-PCR and sequencing will be necessary, such as strain identification and determination of the cleavage site in relation to its level of pathogenicity and epidemiological research.

Progress in the availability of reliable tests to detect the AI virus is essential. The global occurrence of HP H5 in wild migratory and local birds and related outbreaks in poultry and mammals, including the zoonotic risk to humans, has a major societal impact. In addition to the animal suffering caused, there is also an effect on food security and food prices [34]. In areas where H9N2 is endemic, widespread vaccination is carried out to protect the animals. In an increasing number of countries and regions, vaccination against HP H5 is used or considered as an additional tool in the fight against HPAI to prevent and control outbreaks, to reduce economic losses, to lower the risk of human exposure, and to minimize environmental impact [35]. Successful vaccination programs require the implementation of a good monitoring and surveillance system to detect any breakthroughs and, if necessary, to provide direction on what needs to be adjusted to keep the program successful. The tests used within this program must therefore not only be available but also capable of distinguishing infected animals from vaccinated animals (DIVA). Unlike serological tests, both RT-PCR and LAMP tests are fully compatible with all types of AIV vaccines [8]. These tests also meet the growing understanding that testing dead or sick animals is the most efficient way to check both unvaccinated and vaccinated animals for the presence of avian influenza: when dead and sick animals are virus-free, the healthy animals will certainly be [36].

## 5. Conclusions

The Alveo Sense Poultry Avian Influenza Tests represent an advancement in the rapid, on-site detection of avian influenza viruses. These tests, involving cartridge-based RT-LAMP technology and electrical impedance detection, provide a portable, efficient, and reliable method for detecting AIV subtypes H5, H7, and H9 in unprocessed cloacal and oropharyngeal samples. The high specificity and sensitivity demonstrated during the laboratory validation, along with robust reproducibility and stability of the assays, highlight the potential of these tests to enhance early detection in avian influenza outbreaks when using samples of fresh dead or sick birds. The ability to obtain results within 45 min supports the faster decision-making and intervention that are crucial for minimizing the spread of infection and reducing economic impact or public health risks. As field validation continues, the Alveo Sense tests can complement existing diagnostic methods and significantly improve monitoring and surveillance programs, especially in resource-limited settings and areas where suitable laboratories are not available.

## 6. Patents

Alveo’s intellectual property encompasses systems, methods, and diagnostic devices that identify pathogens, genomic materials, proteins, and biomarkers. Alveo Technologies, Inc. currently has four issued U.S. Patents (US 11,465,141; US D927727; US D906526; US D999370) and six patents filed for multiple countries (Pub. 17/416095; Pub. US 17/670193; Pub US63/66460).

## Figures and Tables

**Figure 1 microorganisms-13-01090-f001:**
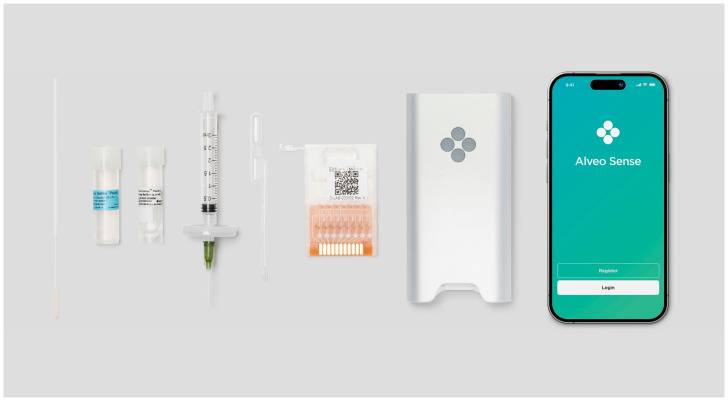
Components of the Alveo Sense Poultry Avian Influenza Test. The test components include the following (**left** to **right**): PurFlock Ultra swab, Alveo Sense Poultry Avian Influenza Swab Elution Solution vial, Alveo Sense Poultry Avian Influenza Assay buffer vial, Alveo Sense Poultry Avian Influenza Syringe Filter, transfer pipette, Alveo Sense Poultry Avian Influenza Type A H5, H7, or H9 cartridge, Alveo Analyzer, and Alveo Sense Mobile Application.

**Figure 2 microorganisms-13-01090-f002:**
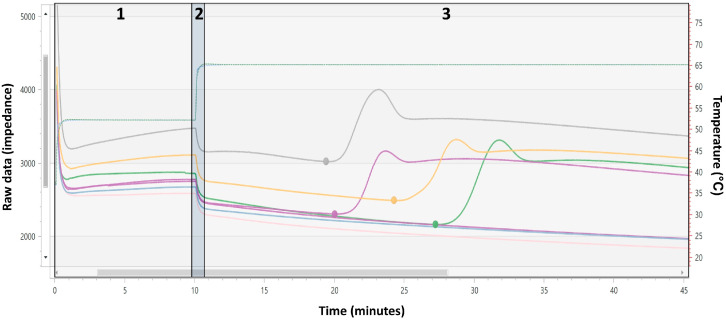
Alveo raw data impedance curve on the left y-axis with overlayed Analyzer temperature (dotted blue and green lines corresponding to right y-axis). Part 1: 52 °C reverse transcription step. Part 2: Transition step from reverse transcription to amplification. Part 3: 65 °C amplification step. Note that the grey, purple, yellow, and green solid lines represent positive amplification curves for the internal control, H5, M target 1, and M target 2, respectively. The two other H5 targets (flat purple and blue lines) and the H7 target (flat pink line) remain negative for this sample.

**Table 1 microorganisms-13-01090-t001:** Technical specificity results of AIV/H5/H7 and AIV/H5/H9 tests from 12 Avian Influenza Virus strains. This table summarizes the qualitative results of AIV/H5/H7 and AIV/H5/H9 testing in oropharyngeal (OP) and cloacal (CL) matrices, demonstrating perfect agreement between expected and observed results. Note that the M and H5 results are summative, as the assays are shared across AIV/H5/H7 and AIV/H5/H9 configurations.

Group	Strain (HN Subtype/Lineage/Clade)	Sample Type	TaqMan PCR (M-Gene) (Ct) *AIV/H5/H7 | AIV/H5/H9*	Qualitative Results (#Positive/#Tested)			
				M	H5	H7	H9
H5 Avian Influenza Viruses	H5N1 (EU_nonGdGD)	OP	30 | 28	6 | 6	6 | 6	0 | 3	0 | 3
		CL	27 | 27	6 | 6	6 | 6	0 | 3	0 | 3
	H5N2 (AM_nonGdGD)	OP	29 | 31	6 | 6	6 | 6	0 | 3	0 | 3
		CL	28 | 28	7 | 7	7 | 7	0 | 4	0 | 3
	H5N2 (EU_nonGdGD)	OP	28 | 29	6 | 6	6 | 6	0 | 3	0 | 3
		CL	25 | 25	6 | 6	6 | 6	0 | 3	0 | 3
	H5N3 (EU_nonGdGD)	OP	28 | 29	7 | 7	7 | 7	0 | 4	0 | 3
		CL	26 | 25	6 | 6	6 | 6	0 | 3	0 | 3
	H5N3 (EU_nonGdGD)	OP	29 | 28	6 | 6	6 | 6	0 | 3	0 | 3
		CL	26 | 26	6 | 6	6 | 6	0 | 3	0 | 3
	H5N8 (2.3.4.4b)	OP	31 | 25	6 | 6	6 | 6	0 | 3	0 | 3
		CL	27 | 24	6 | 6	6 | 6	0 | 3	0 | 3
Non-H5 Avian Influenza Viruses	H3N1	OP	28 | 32	8 | 8	0 | 8	0 | 4	0 | 4
		CL	25 | 25	6 | 6	0 | 6	0 | 3	0 | 3
	H6N1	OP	28 | 30	8 | 8	0 | 8	0 | 4	0 | 4
		CL	26 | 26	6 | 6	0 | 6	0 | 3	0 | 3
	H7N1 (Eurasian lineage)	OP	29 | 31	6 | 6	0 | 6	3 | 3	0 | 3
		CL	26 | 26	6 | 6	0 | 6	3 | 3	0 | 3
	H7N7 (Eurasian lineage)	OP	29 | 31	6 | 6	0 | 6	3 | 3	0 | 3
		CL	27 | 24	6 | 6	0 | 6	3 | 3	0 | 3
	H9N2 (G lineage)	OP	28 | 29	6 | 6	0 | 6	0 | 3	3 | 3
		CL	25 | 25	6 | 6	0 | 6	0 | 3	3 | 3
	H9N2 (G lineage)	OP	26 | 28	7 | 7	0 | 7	0 | 4	3 | 3
		CL	26 | 26	6 | 6	0 | 6	0 | 3	3 |3

**Table 2 microorganisms-13-01090-t002:** Technical specificity results of AIV/H5/H7 and AIV/H5/H9 tests from 19 Non-Avian Influenza Virus Microbial Organisms. This table summarizes the qualitative results from AIV/H5/H7 and AIV/H5/H9 testing of nucleic acid isolates from various non-AIV pathogens, demonstrating perfect agreement between expected and observed results. Note that the M and H5 results are summative, as the assays are shared across AIV/H5/H7 and AIV/H5/H9 configurations.

Group	Strain (Serotype or Subtype)	TaqMan PCR (Ct) *AIV/H5/H7 | AIV/H5/H9*	Qualitative Results (#Positive/#Tested)			
			M	H5	H7	H9
Non-AIV Pathogens	Infectious bronchitis virus, D388_GI-13	22 | 22	0 | 4	0 | 4	0 | 2	0 | 2
	Infectious bronchitis virus, D274_GI-12	21 | 21	0 | 4	0 | 4	0 | 2	0 | 2
	Infectious bronchitis virus, M41_GI-1	23 | 23	0 | 4	0 | 4	0 | 2	0 | 2
	Infectious bronchitis virus, 4/91_GI-13	27 | 27	0 | 4	0 | 4	0 | 2	0 | 2
	Infectious laryngotracheitis virus	24 | 24	0 | 4	0 | 4	0 | 2	0 | 2
	Avian metapneumovirus (AMPV-A)	25 | 26	0 | 4	0 | 4	0 | 2	0 | 2
	Avian metapneumovirus (AMPV-B)	21 | 23	0 | 4	0 | 4	0 | 2	0 | 2
	Avian metapneumovirus (AMPV-C)	27 | 27	0 | 4	0 | 4	0 | 2	0 | 2
	Newcastle Disease virus (NDV)	22 | 22	0 | 4	0 | 4	0 | 2	0 | 2
	*Mycoplasma synoviae*	21 | 21	0 | 4	0 | 4	0 | 2	0 | 2
	*Mycoplasma synoviae*	26 | 26	0 | 4	0 | 4	0 | 2	0 | 2
	*Mycoplasma gallisepticum*	22 | 22	0 | 4	0 | 4	0 | 2	0 | 2
	*Avibacterium paragallinarum A1*	23 | 23	0 | 4	0 | 4	0 | 2	0 | 2
	*Avibacterium paragallinarum B1*	23 | 23	0 | 4	0 | 4	0 | 2	0 | 2
	*Avibacterium paragallinarum C4*	24 | 24	0 | 4	0 | 4	0 | 2	0 | 2
	*Pasteurella multocida*	24 | 24	0 | 4	0 | 4	0 | 2	0 | 2
	*Gallibacterium anatis*	22 | 22	0 | 4	0 | 4	0 | 2	0 | 2
	*Ornithobacterium rhinotracheale (ORT)*	- *	0 | 4	0 | 4	0 | 2	0 | 2
	*Riemerella anatipestifer*	- *	0 | 4	0 | 4	0 | 2	0 | 2

* No TaqMan PCR results available, strong positive cultures of >4 McFarland.

**Table 3 microorganisms-13-01090-t003:** Results for the Alveo Sense AIV/H5/H7 Test. This table summarizes the results from AIV spiked sample testing of H5N2, H5N8, and H7N7 across different oropharyngeal (OP) and cloacal (CL) matrices and pool sizes.

Strain (Subtype/Lineage/Clade)	Sample Type	TaqMan PCR (M-Gene) (Ct)	Qualitative Results (#Positive | #Tested)		
			M	H5	H7
H5N2 (AM_nonGdGD) individual	OP	33	2 | 2	2 | 2	0 | 2
	CL	29	2 | 2	2 | 2	0 | 2
H5N2 (AM_nonGdGD) pools of 3	OP	34	2 | 2	2 | 2	0 | 2
	CL	29	2 | 2	2 | 2	0 | 2
H5N2 (AM_nonGdGD) pools of 5	OP	34	2 | 2	2 | 2	0 | 2
	CL	35	2 | 2	2 | 2	0 | 2
H5N2 (AM_nonGdGD) pools of 10	OP	36	2 | 2	1 | 2	0 | 2
H5N8 (2.3.4.4b) individual	OP	28	2 | 2	2 | 2	0 | 2
	CL	30	2 | 2	2 | 2	0 | 2
H5N8 (2.3.4.4b) pools of 3	OP	28	2 | 2	2 | 2	0 | 2
	CL	29	2 | 2	2 | 2	0 | 2
H5N8 (2.3.4.4b) pools of 5	OP	30	2 | 2	2 | 2	0 | 2
	CL	30	2 | 3	3 | 3	0 | 3
H5N8 (2.3.4.4b) pools of 10	OP	31	3 | 3	3 | 3	0 | 3
H7N7 (Eurasian lineage) individual	OP	24	2 | 2	0 | 2	2 | 2
	CL	28	2 | 2	0 | 2	2 | 2
H7N7 (Eurasian lineage) pools of 3	OP	25	2 | 2	0 | 2	2 | 2
	CL	30	2 | 2	0 | 2	2 | 2
H7N7 (Eurasian lineage) pools of 5	OP	25	2 | 2	0 | 2	2 | 2
	CL	27	3 | 3	0 | 3	3 | 3
H7N7 (Eurasian lineage) pools of 10	OP	28	3 | 3	0 | 3	3 | 3

**Table 4 microorganisms-13-01090-t004:** Results for the Alveo Sense AIV/H5/H9 test. This table summarizes the results from AIV spiked sample testing of H5N2, H5N8, and H9N2 across different oropharyngeal (OP) and cloacal (CL) matrices and pool sizes.

Strain (Subtype/Lineage/Clade)	Sample Type	TaqMan PCR (M-Gene) (Ct)	Qualitative Results (#Positive | #Tested)		
			M	H5	H9
H5N2 (AM_nonGdGD) individual	OP	31	2 | 2	2 | 2	0 | 2
	CL	30	2 | 2	2 | 2	0 | 2
H5N2 (AM_nonGdGD) pools of 3	OP	33	2 | 2	2 | 2	0 | 2
	CL	30	2 | 2	2 | 2	0 | 2
H5N2 (AM_nonGdGD) pools of 5	OP	31	2 | 2	2 | 2	0 | 2
	CL	31	2 | 2	2 | 2	0 | 2
H5N2 (AM_nonGdGD) pools of 10	OP	30	3 | 3	3 | 3	0 | 3
H5N8 (2.3.4.4b) individual	OP	29	2 | 2	2 | 2	0 | 2
	CL	29	2 | 2	2 | 2	0 | 2
H5N8 (2.3.4.4b) pools of 3	OP	30	2 | 2	2 | 2	0 | 2
	CL	29	2 | 2	2 | 2	0 | 2
H5N8 (2.3.4.4b) pools of 5	OP	30	2 | 2	2 | 2	0 | 2
	CL	28	2 | 2	2 | 2	0 | 2
H5N8 (2.3.4.4b) pools of 10	OP	28	3 | 3	3 | 3	0 | 3
H9N2 (G lineage) individual	OP	26	2 | 2	0 | 2	2 | 2
	CL	28	2 | 2	0 | 2	2 | 2
H9N2 (G lineage) pools of 3	OP	27	2 | 2	0 | 2	2 | 2
	CL	29	2 | 2	0 | 2	2 | 2
H9N2 (G lineage) pools of 5	OP	28	2 | 2	0 | 2	2 | 2
	CL	29	2 | 2	0 | 2	2 | 2
H9N2 (G lineage) pools of 10	OP	28	3 | 3	0 | 3	3 | 3

## Data Availability

The original contributions presented in this study are included in the article, further inquiries can be directed to the corresponding author.

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
