# Peer review of "Development and Laboratory Validation of Rapid, Bird-Side Molecular Diagnostic Assays for Avian Influenza Virus Including Panzootic H5Nx"

_microorganisms, 2025, doi:10.3390/microorganisms13051090_

Round 1

Reviewer 1 Report (Previous Reviewer 1)

Comments and Suggestions for Authors

The authors addressed satisfactorely the original concerns.

Author Response

As reviewer 1 had no comments: no reply

Reviewer 2 Report (Previous Reviewer 2)

Comments and Suggestions for Authors

This study introduces the Alveo Sense Poultry Avian Influenza Tests, a rapid, on-site diagnostic tool using RT-LAMP and impedance-based detection to identify avian influenza virus (AIV) subtypes H5, H7, and H9 in unprocessed cloacal and oropharyngeal samples. The tests, which deliver results within 45 minutes, demonstrated high specificity, sensitivity, and robustness in laboratory validation, with no cross-reactivity with non-AIV pathogens. The authors highlight the potential of these tests for early outbreak detection and control, particularly in resource-limited settings. However, field validation is needed to confirm diagnostic performance across diverse avian species and real-world conditions. Below are my comments and suggestions:

  • Why were only two M gene assays selected despite testing 12 designs? What criteria excluded the others?
  • For the H5 assay, three primer sets were necessary due to genetic diversity. Could this complexity lead to variability in field performance, especially with emerging strains?
  • The LoD95 ranges (Ct 29–33) are acceptable for symptomatic birds but may miss low viral loads in asymptomatic cases. How do the authors propose addressing this limitation in surveillance contexts?
  • The manuscript emphasizes testing "fresh dead or sick birds." How would the assay perform in healthy birds during routine surveillance, where viral loads are lower?
  • The spiked sample data is promising, but how representative are these of real-world field samples, which may contain inhibitors or degraded RNA?
  • The authors compare LoD to RT-PCR and lateral flow tests but omit discussion of cost, ease of use, or training requirements. How does Alveo Sense balance accuracy with accessibility for non-specialists? How does the assay’s 45-minute turnaround compare to newer portable systems (e.g., RPA or LAMP)?
  • Did the authors test how might environmental factors (e.g., temperature fluctuations, sample handling errors) impact field performance?
  • For the cartridge’s closed system, what is the shelf life under field conditions (e.g., tropical climates)?
  • For regions considering H5 vaccination, how does this test align with DIVA (Differentiating Infected from Vaccinated Animals) strategies?
  • The assay’s utility for mammals (mentioned briefly) is intriguing. Were any mammalian samples evaluated, given H5N1’s zoonotic spread?
  • How might this assay integrate with existing global surveillance networks (e.g., WOAH, FAO)?
  • Page 2: "neutralinidase" should be "neuraminidase."
  • Page 11: Table 1’s header "TagMan PCR (Ct)", please specify if it’s for M or subtype targets.
Comments on the Quality of English Language

The English could be improved to more clearly express the research.

Author Response

Comments and Suggestions – Microorganisms, Updated on 23-April-2025

Reviewer 2:

This study introduces the Alveo Sense Poultry Avian Influenza Tests, a rapid, on-site diagnostic tool using RT-LAMP and impedance-based detection to identify avian influenza virus (AIV) subtypes H5, H7, and H9 in unprocessed cloacal and oropharyngeal samples. The tests, which deliver results within 45 minutes, demonstrated high specificity, sensitivity, and robustness in laboratory validation, with no cross-reactivity with non-AIV pathogens. The authors highlight the potential of these tests for early outbreak detection and control, particularly in resource-limited settings. However, field validation is needed to confirm diagnostic performance across diverse avian species and real-world conditions. Below are my comments and suggestions:

Q: Why were only two M gene assays selected despite testing 12 designs? What criteria excluded the others?

A: As part of the development process, we designed and tested multiple primer sets to identify those that offered the highest sensitivity and specificity, while also complementing each other to optimize overall assay performance. Ultimately, two designs were selected based on their strong sensitivity, successful detection of all available Avian Influenza virus strains, and low risk of false positives. No modifications were made to the manuscript, as the rationale for primer selection is detailed in lines 323 to 326.

‘Based on technical performance and broad detection capabilities, two assays were ultimately selected to maximize specificity. These two assays target different regions of the M gene and complement each other in coverage, thereby reducing the risk of false negative results due to primer mismatches.’

Q: For the H5 assay, three primer sets were necessary due to genetic diversity. Could this complexity lead to variability in field performance, especially with emerging strains?

A: Thank you for this comment—indeed, the complexity of H5 is substantial. We developed our assays to cover a broad range of relevant lineages, successfully detecting all available Avian Influenza virus strains in our wet-lab testing, while also demonstrating strong homology with existing strains, as described in the manuscript. By designing three assays specifically targeting H5, our aim was to account for sufficient genetic variability to detect both existing and potentially emerging strains. This is particularly evident in the development of two distinct assays capable of targeting the 2.3.4.4b clade.

However, as with any molecular-based diagnostic, including RT-PCR, the emergence of new strains—especially in highly variable regions—can impact assay performance. This underscores the importance of continuous surveillance of circulating strains, as emphasized in lines 541–544 of the manuscript: 'These findings show that consistent monitoring of the reported sequences of the M and H genes is needed to ensure that the selected primers and probes used in genomic tests such as RT-PCR and LAMP assays are capable of detecting all relevant strains.'

As this explanation is already included in the manuscript, no modifications have been made.

Q: The LoD95 ranges (Ct 29–33) are acceptable for symptomatic birds but may miss low viral loads in asymptomatic cases. How do the authors propose addressing this limitation in surveillance contexts?

A: As noted in the manuscript, proper sampling and the use of complementary diagnostic methods remain important, as we do not claim our test to be a one-size-fits-all solution. We believe that additional field testing is necessary to fully evaluate the test’s applicability for surveillance and practical use. A detailed description of this is already provided in lines 581–590 of the manuscript.

‘The first results of the Alveo Sense test on the field samples collected in the acute phase of the LP H5N1 outbreak give confidence, but the dataset needs further expansion. When expanding the number of field data, it is also of importance to include samples from other animal species such as turkeys, ducks, geese, and mammals in order to know what the test can be used for with high reliability. Depending on the results and the requirements for diagnosing AIV infection, it can then be decided where the test is best used, either as a screening test or as a decisive test. In the event of various questions and circumstances, the further use of confirmatory tests such as RT-PCR and sequencing will be necessary, such as strain identification, determination of the cleavage site in relation to its level of pathogenicity and epidemiological research.’

Q:The manuscript emphasizes testing "fresh dead or sick birds." How would the assay perform in healthy birds during routine surveillance, where viral loads are lower?

A: Similar to our response to the previous comment, we believe that additional field testing is necessary to fully assess the test’s real-world applicability for surveillance and use, which could serve as the basis for a valuable follow-up publication.

Q: The spiked sample data is promising, but how representative are these of real-world field samples, which may contain inhibitors or degraded RNA?

A: Thank you for this point. As described in the manuscript, all tests were conducted using cloacal or oropharyngeal matrix collected from SPF chickens to simulate the presence of inhibitors and RNases typically found in real-world field samples. To further clarify this, we have added the following sentence to lines 197-198 of the manuscript:

The collected matrix was used during testing to simulate the inhibitors found in real-world field samples.’

Q: The authors compare LoD to RT-PCR and lateral flow tests but omit discussion of cost, ease of use, or training requirements. How does Alveo Sense balance accuracy with accessibility for non-specialists? How does the assay’s 45-minute turnaround compare to newer portable systems (e.g., RPA or LAMP)?

A: The Alveo Sense mobile application includes step-by-step instructions that guide users through the entire workflow, making the test accessible and easy to use with minimal training. To emphasize this, we have added the sentence, 'the mobile application’s step-by-step instructions make testing easy for new and experienced users,' to lines 518–519 of the manuscript.

Since test results are generated by the Alveo Analyzer using a complex algorithm and presented to the end user as Negative, Positive, or Invalid, there is no subjective interpretation required. As such, test accuracy is not dependent on user experience. This is already described in the manuscript (lines 518–523), and therefore no additional modifications were made in response to this point.

Regarding the turnaround times of newer portable systems, we do not have sufficient information to comment.

Q: Did the authors test how might environmental factors (e.g., temperature fluctuations, sample handling errors) impact field performance?

A: Thank you for this comment. We assessed post-rehydration reagent stability by evaluating the impact of time delays between reagent rehydration and test initiation. This allowed us to examine the effect of operator delay on performance, with the results detailed in Section 3.6. Our findings indicate that the test demonstrates strong robustness and time tolerance, accommodating variability in this aspect.

However, we did not assess the impact of external temperature fluctuations in the current study. Future testing is planned to evaluate performance across different environmental conditions, and the findings from these studies could be presented in a follow-up publication.

While it is not feasible to account for every potential field condition, each test cartridge includes an internal process control that would yield an invalid result in the case of improper sample handling or abnormal testing conditions. This process control is referenced in the Abstract (line 25) and the Materials and Methods section (lines 125–126).

To clarify this point further, we have added the sentence, 'While our findings indicate strong robustness, the process control embedded in each cartridge would yield an invalid result in the event of major operator errors or sample-related issues,' to the end of Section 3.6 (lines 450–452).

Q: For the cartridge’s closed system, what is the shelf life under field conditions (e.g., tropical climates)

A: The Alveo Sense cartridge is stored in an air-tight pouch with desiccant and has a storage condition of 2°C to 30°C, with a current stability claim of 12 months. While accelerated stability testing has been conducted at elevated temperatures, including 37°C and 45°C, no long-term stability claims have been made for these conditions, and it is not recommended to store the cartridges above 30°C. To enhance clarity, we have updated the pre-rehydration shelf life statement to include the storage conditions. The revised sentence in lines 281–283 now reads:

'The pre-rehydration shelf life of the AIV cartridge with dried-down reagents is currently established at 12 months at a storage condition of 2–30°C (similar to the ALVEO be.well COVID-19 assay, data not shown), with ongoing stability studies in progress.'

Q: For regions considering H5 vaccination, how does this test align with DIVA (Differentiating Infected from Vaccinated Animals) strategies?

A: The Alveo Sense Avian Influenza test facilitates a DIVA vaccination strategy when used in combination with the administration of inactivated avian influenza vaccines. Inactivated vaccine viruses do not replicate in the host and hence detection of avian influenza RNA by Alveo’s portable LAMP technology platform in oropharyngeal or cloacal sampling sites can only signify the presence of a field virus. Successful application of inactivated avian influenza vaccines in the absence of field virus infection will yield a negative result. In this way, the avian influenza status of a flock can be clearly determined. No modification was made to the manuscript, as this message is already described in the Discussion (lines 603-605):

‘Unlike serological tests, both RT-PCR and LAMP tests are fully compatible with all types of AIV vaccines [8].’

Q: The assay’s utility for mammals (mentioned briefly) is intriguing. Were any mammalian samples evaluated, given H5N1’s zoonotic spread?

A: While we agree that this is an intriguing point, we did not evaluate any mammals during the development and validation of the Alveo Sense test. This presents an excellent opportunity for a follow-up study and publication to explore the test's performance across different mammalian species.

Q: How might this assay integrate with existing global surveillance networks (e.g., WOAH, FAO)?

A: This is an excellent point that we believe can be expanded upon. In addition to the pen-side test, the Alveo Sense platform includes a software program, Alveo Vista, which provides real-time access to up-to-date information on potential problem areas through time-stamped, geo-tagged results. All data is stored in a secure, encrypted portal and allows for varying levels of access within an organization. A platform like this could complement existing global surveillance networks by providing real-time insights into outbreaks in specific regions. To highlight this, we have added the following information to the manuscript in lines 524–529:

'In addition to the pen-side test results, the Alveo Sense platform includes a secure results database—Alveo Vista—that provides real-time access to up-to-date information through time-stamped and geo-tagged results. All data is stored in an encrypted portal, which can support varying levels of access to organizations and public health officials. This combination of early testing and centralized data monitoring can complement existing global surveillance networks and enhance monitoring programs.'

Q: Page 2: "neutralinidase" should be "neuraminidase."

A: We do not see any instances of “neutralinidase” in the manuscript. No modifications have been made.

Q: Page 11: Table 1’s header "TagMan PCR (Ct)", please specify if it’s for M or subtype targets.

A: Thank you for this point of clarification. The Avian Influenza TaqMan PCR used in this study targets the M-gene. To address this, we have specified the M-gene target for the TaqMan PCR assay in Tables 1, 3, and 4.

Reviewer 3 Report (Previous Reviewer 3)

Comments and Suggestions for Authors

The reviewed manuscript is dedicated to the design and validation of an RT-LAMP-based portable device for detection of several Avian influenza virus subtypes (H5, H7, H9). The manuscript itself is well-written and the test is validated enough for application in a routine veterinary practice.

Author Response

as Reviewer 3 had no comments: no reply

This manuscript is a resubmission of an earlier submission. The following is a list of the peer review reports and author responses from that submission.

Round 1

Reviewer 1 Report

Comments and Suggestions for Authors

The authors developed a molecular rapid test based on LAMP to detect the M gene of Influenza and the H5, H7 and H9 hemagglutinin genes. Some concerns should be addressed before acceptance of this manuscript.

  1. The asterisk of the corresponding author is lacking.
  2. Introduction: the reason of the high pathogenicity acquisition should be better described (furin cleavage site). This paper may be useful: Lee DH, et al. Genome sequences of haemagglutinin cleavage site predict the pathogenicity phenotype of avian influenza virus: statistically validated data for facilitating rapid declarations and reducing reliance on in vivo testing. Avian Pathol. 2024 Aug;53(4):242-246. doi: 10.1080/03079457.2024.2317430.
  3. Abstract and Table 1. The authors refer to technical specificity without mentioning the (high) sensitivity of the test.
  4. Why two test were developed instead of one single A H5 H7 H9?
  5. I am not clear about Tables 1 and 2. I understand that the qualitative results shown are from the tests developed in this study. So why are these tables in Materials and Methods?
  6. The methodology should be explained with some more details. For example, at what temperature is performed the LAMP assay? Is there a need for any (small) additional equipment?
  7. The limit of detection of the assay is also reported in Materials and Methods, while this is an important result. The authors should avoid also duplicate information between Methods and Results.
  8. The limit of detection seems to be adequate, although lower in terms of viral load, than qRT-PCR. How this performance compares with antigenic assays?

Author Response

Comment 1 The asterisk of the corresponding author is lacking.

ANSWER: The asterisk has been added (page 1, line 7)

Comment 2: Introduction: the reason of the high pathogenicity acquisition should be better described (furin cleavage site). This paper may be useful: Lee DH, et al. Genome sequences of haemagglutinin cleavage site predict the pathogenicity phenotype of avian influenza virus: statistically validated data for facilitating rapid declarations and reducing reliance on in vivo testing. Avian Pathol. 2024 Aug;53(4):242-246. 

ANSWER: We agree and have added the following sentences to the introduction (page 2, lines 47-56) that includes the reference that was suggested: ‘The number of basic amino acids in the HA0 cleavage site plays a critical role in pathogenicity, determining which proteases can cleave HA and in which tissues the AIV can replicate [2]. LPAIV usually have a cleavage site that contains less than two basic amino acids in a critical position and are therefore cleaved by trypsin or trypsin-like proteases,  limiting replication of these viruses, principally, to epithelial cells of the intestinal and respiratory tracts of birds. By contrast, multi-basic cleavage site contains several basic amino acids at the same critical position and is cleaved by several common cellular proteases  present in most cells throughout the body, causing systemic disease and lethal infection of high pathogenicity AIV (HPAIV) in gallinaceous poultry species [2].’

Comment 3: Abstract and Table 1. The authors refer to technical specificity without mentioning the (high) sensitivity of the test.

ANSWER: Thank you, we have added ‘diagnostic sensitivity’ to the abstract (page 1, line 26-27).

Comment 4: Why two test were developed instead of one single A H5 H7 H9?

ANSWER: A single test would require 8 channels which are not available on the cartridge. For this reason, 2 tests were developed. The H9 containing product was developed specifically for geographical regions where this virus is known to be endemic. 

Comment 5: I am not clear about Tables 1 and 2. I understand that the qualitative results shown are from the tests developed in this study. So why are these tables in Materials and Methods?

ANSWER: We have moved Table 1 and Table 2 to the Results section as suggested by the reviewer.

Comment 6: The methodology should be explained with some more details. For example, at what temperature is performed the LAMP assay? Is there a need for any (small) additional equipment? 

ANSWER: We added more information about the temperature methodology in lines 134-136 (page 3) and in Figure 2 (page 4). There is no need for additional equipment, as all required equipment is shown in Figure 1 (page 4).

Comment 7: The limit of detection of the assay is also reported in Materials and Methods. While this is an important result, the authors should avoid also duplicate information between Methods and Results.

ANSWER: We think this is a misunderstanding. In the Materials section, we describe the way we determined the LoD with samples tested by RT-PCR. In the Results section, we present the performance of the Alveo Sense test on the same samples. Therefore, we did not adjust the text.

Comment 8: The limit of detection seems to be adequate, although lower in terms of viral load, than qRT-PCR. How this performance compares with antigenic assays?

ANSWER: In this paper, we report on the development and laboratory validation of our test. We did not make direct comparisons with antigenic tests, and thus, would not like to speculate too much about performance comparisons. We added lines 525 to 527 (page 14) and reference 19 (Soliman et al, 2010) to the Discussion, that in conjunction with paper 18 (Slomka et al, lines 522-525 (page 14)) suggest the LFDs would be less sensitive. However, we prefer not to highlight the expected differences as this is not the goal of our paper.

Reviewer 2 Report

Comments and Suggestions for Authors

The manuscript presents the development and validation of the Alveo Sense Poultry Avian Influenza Tests, which aim to provide rapid, on-site detection of avian influenza virus (AIV) subtypes H5, H7, and H9 using reverse-transcription loop-mediated isothermal amplification (RT-LAMP) and impedance-based measurements. While the study addresses an important need for rapid diagnostics in avian influenza outbreaks, there are several scientific concerns and weaknesses. Below are the key issues:

  • The manuscript heavily relies on laboratory validation using spiked samples and archived strains. While the authors mention that field validation is ongoing, the absence of extensive real-world field data significantly weakens the study's claims about the test's applicability in actual outbreak scenarios. How do the authors plan to address the potential variability in field conditions, such as differences in sample quality, environmental factors, and operator expertise, which could affect the test's performance? Without robust field validation, the test's reliability in real-world settings remains unproven. The manuscript should be rejected until comprehensive field data is available to support the claims.
  • The study claims broad detection capabilities for H5, H7, and H9 subtypes, but the validation appears to be limited to a relatively small number of strains (12 AIV strains and 19 non-AIV pathogens). Given the high genetic variability of AIV, this limited panel may not adequately represent the diversity of circulating strains. How do the authors ensure that the selected primer sets will detect all relevant strains, especially given the rapid evolution and genetic drift of AIV? The manuscript should include a more extensive validation panel, encompassing a wider range of geographically and temporally diverse strains, to demonstrate the test's robustness against genetic variability.
  • The study reports a decrease in sensitivity at higher Ct values (e.g., Ct > 30), which is a critical limitation for a diagnostic test intended for early detection. The authors acknowledge that the test's sensitivity drops as viral load decreases, which could lead to false negatives in subclinical or early-stage infections. How do the authors justify the test's utility in early outbreak detection when its sensitivity is compromised at low viral loads, which are often present in the initial stages of infection? The test's inability to reliably detect low viral loads undermines its potential as a rapid diagnostic tool for early intervention. This limitation should be addressed before the manuscript can be considered for publication.
  • The manuscript lacks a thorough comparison with existing diagnostic methods, such as RT-PCR and lateral flow devices (LFDs). While the authors briefly mention the advantages of RT-LAMP over RT-PCR, they do not provide a detailed performance comparison, particularly in terms of sensitivity, specificity, and turnaround time. How does the Alveo Sense test compare to RT-PCR and LFDs in terms of diagnostic sensitivity, specificity, and practical utility in resource-limited settings? A more rigorous comparison with established methods is necessary to demonstrate the test's superiority or equivalence. Without this, the manuscript's claims about the test's advantages remain unsubstantiated.
  • While the authors report 100% specificity in their validation studies, the limited panel of non-AIV pathogens tested (19 organisms) raises concerns about potential cross-reactivity with other avian pathogens not included in the study. Have the authors tested the assay against a broader range of avian pathogens, particularly those that are commonly co-circulating with AIV in poultry populations? The risk of cross-reactivity and false positives must be thoroughly investigated, especially given the high stakes of an avian influenza diagnosis. The manuscript should be rejected until a more comprehensive specificity study is conducted.
  • The manuscript mentions the ability to test pooled samples (up to 10 oropharyngeal swabs or 5 cloacal swabs), but the data supporting this claim is limited. Pooling samples can dilute viral load, potentially reducing the test's sensitivity. How does the test perform when detecting low viral loads in pooled samples, and what is the impact of pooling on the test's sensitivity and specificity? The manuscript should provide more detailed data on pooled sample testing, including the impact of pooling on detection limits and the risk of false negatives.
  • The study briefly mentions post-rehydration reagent stability but does not provide sufficient data on the long-term stability of the reagents or the shelf life of the test cartridges. This information is critical for assessing the test's practicality in field settings. What is the shelf life of the test cartridges, and how stable are the reagents under various storage conditions (e.g., temperature, humidity)? Without clear data on reagent stability and shelf life, the test's suitability for use in resource-limited or remote settings cannot be adequately evaluated.
  • The manuscript makes strong claims about the test's utility for early detection and outbreak control, but the data presented do not fully support these claims. The limitations in sensitivity, lack of field validation, and insufficient comparison with existing methods undermine the authors' conclusions. How do the authors reconcile the test's limitations with their claims about its potential impact on avian influenza control? The manuscript's conclusions are overly optimistic given the data presented. The authors should temper their claims and provide a more balanced discussion of the test's limitations.
Comments on the Quality of English Language

The English could be improved to more clearly express the research.

Author Response

The manuscript presents the development and validation of the Alveo Sense Poultry Avian Influenza Tests, which aim to provide rapid, on-site detection of avian influenza virus (AIV) subtypes H5, H7, and H9 using reverse-transcription loop-mediated isothermal amplification (RT-LAMP) and impedance-based measurements. While the study addresses an important need for rapid diagnostics in avian influenza outbreaks, there are several scientific concerns and weaknesses. Below are the key issues:

Comment 1: The manuscript heavily relies on laboratory validation using spiked samples and archived strains. While the authors mention that field validation is ongoing, the absence of extensive real-world field data significantly weakens the study's claims about the test's applicability in actual outbreak scenarios. How do the authors plan to address the potential variability in field conditions, such as differences in sample quality, environmental factors, and operator expertise, which could affect the test's performance? Without robust field validation, the test's reliability in real-world settings remains unproven. The manuscript should be rejected until comprehensive field data is available to support the claims. 

ANSWER: We do not agree that we claim that the test will perform well in the field.  As stated in the original manuscript at several places (line 107 (page 3) of the Introduction, heading of paragraph 2.5 (page 6, line 229)), this paper reports the development and laboratory validation of the Alveo Sense test. We also mention in the abstract, discussion, and conclusion, that a field validation is required to see how the test performs in the field. To further clarify that we have not yet validated the test under field conditions, we added the word ‘laboratory’ to the Title, the Abstract (page 1, line 26), and the Conclusion (page 15, line 584).

Comment 2: The study claims broad detection capabilities for H5, H7, and H9 subtypes, but the validation appears to be limited to a relatively small number of strains (12 AIV strains and 19 non-AIV pathogens). Given the high genetic variability of AIV, this limited panel may not adequately represent the diversity of circulating strains. How do the authors ensure that the selected primer sets will detect all relevant strains, especially given the rapid evolution and genetic drift of AIV? The manuscript should include a more extensive validation panel, encompassing a wider range of geographically and temporally diverse strains, to demonstrate the test's robustness against genetic variability. 

ANSWER: The validation of the Alveo Sense tests for the broad detection of AIV was performed in two ways: the wet-lab panel as mentioned by reviewer 2, and the complementary in silico study using almost 10,000 sequences as described in paragraph 3.1 (page 8, lines 326-351). This approach warrants our broad detection claim which is not challenged by reviewers 1 and 3. Despite these results, we acknowledge the risk of missing new strains of AIV due to the genetic evolution, as already stated in lines 513-515 (page 14). The check for the specificity included not only wet-lab testing of 19 of the most prevalent poultry pathogens, but also a primer BLAST against the NCBI nucleotide database. By combining approaches, we ensured that no cross-reactivity with non-target sequences was possible (page 5, lines 172-174). In conclusion, we did not amend the manuscript in response to this comment of reviewer 2 as the combination of the laboratory testing, extensive in silico testing, and our discussion about the need for consistent monitoring of the reported sequences of the M and H genes to ensure that the selected primers are capable of detecting all relevant strains (page 14, lines 505-508) warrants our claim of broad detection.

Comment 3: The study reports a decrease in sensitivity at higher Ct values (e.g., Ct > 30), which is a critical limitation for a diagnostic test intended for early detection. The authors acknowledge that the test's sensitivity drops as viral load decreases, which could lead to false negatives in subclinical or early-stage infections. How do the authors justify the test's utility in early outbreak detection when its sensitivity is compromised at low viral loads, which are often present in the initial stages of infection? The test's inability to reliably detect low viral loads undermines its potential as a rapid diagnostic tool for early intervention. This limitation should be addressed before the manuscript can be considered for publication.  

ANSWER: We fully agree that when the bird is in its early stage of infection, the viral load might be low for the developed test. For this reason, we already mentioned the importance of testing fresh dead of sick birds (see lines 101, 529, 544, 548 and 573). These birds do not die in first phase of infection and hence act as the red flags for early phase flock infection. To further clarify this for the reader, we have added ‘on samples from fresh dead and sick birds’ and ‘flock-level’ to line 35 (page 1) of the Abstract, and ‘when using samples of fresh dead or sick birds’ to line 586 (page 15) of the Conclusion.

Comment 4: The manuscript lacks a thorough comparison with existing diagnostic methods, such as RT-PCR and lateral flow devices (LFDs). While the authors briefly mention the advantages of RT-LAMP over RT-PCR, they do not provide a detailed performance comparison, particularly in terms of sensitivity, specificity, and turnaround time. How does the Alveo Sense test compare to RT-PCR and LFDs in terms of diagnostic sensitivity, specificity, and practical utility in resource-limited settings? A more rigorous comparison with established methods is necessary to demonstrate the test's superiority or equivalence. Without this, the manuscript's claims about the test's advantages remain unsubstantiated. 

ANSWER: Our paper reports on the development and laboratory validation of the Alveo Sense tests. As the gold standard RT-PCR was used, a thorough comparison of all aspects of all diagnostics tests was not the purpose of our research and would require a second or third paper to fully explore. We presented a review (paper 9, Feddema et al) regarding the advantages and disadvantages of LAMP based tests to briefly discuss the general aspects. For these reasons, we did not amend the text of the manuscript.

Comment 5: While the authors report 100% specificity in their validation studies, the limited panel of non-AIV pathogens tested (19 organisms) raises concerns about potential cross-reactivity with other avian pathogens not included in the study. Have the authors tested the assay against a broader range of avian pathogens, particularly those that are commonly co-circulating with AIV in poultry populations? The risk of cross-reactivity and false positives must be thoroughly investigated, especially given the high stakes of an avian influenza diagnosis. The manuscript should be rejected until a more comprehensive specificity study is conducted. 

ANSWER: This is not correct – the panel for testing of 19 relevant non-AIV pathogens was chosen to be broad but never intended to be all inclusive (impossible in any case). The check for specificity did not only include wet-lab testing of 19 highly prevalent poultry pathogens, but also included a primer BLAST against the NCBI nucleotide database to ensure no cross-reactivity with non-target sequences, as mentioned on lines 172-174 (page 5). We did not amend the manuscript.

Comment 6: The manuscript mentions the ability to test pooled samples (up to 10 oropharyngeal swabs or 5 cloacal swabs), but the data supporting this claim is limited. Pooling samples can dilute viral load, potentially reducing the test's sensitivity. How does the test perform when detecting low viral loads in pooled samples, and what is the impact of pooling on the test's sensitivity and specificity? The manuscript should provide more detailed data on pooled sample testing, including the impact of pooling on detection limits and the risk of false negatives. 

ANSWER: Our results discuss the Limits of detection which are independent of the level of pooling. How many samples can be pooled in the field depends heavily on the selected birds. As we advise using samples of fresh dead or sick birds, the quantity of viruses will be substantial when these deaths are caused by AIV. We did not amend the manuscript regarding the development and laboratory validation in response to this comment. A second, subsequent paper as a following to the current validation would be a suitable place to discuss such field results.

Comment 7: The study briefly mentions post-rehydration reagent stability but does not provide sufficient data on the long-term stability of the reagents or the shelf life of the test cartridges. This information is critical for assessing the test's practicality in field settings. What is the shelf life of the test cartridges, and how stable are the reagents under various storage conditions (e.g., temperature, humidity)? Without clear data on reagent stability and shelf life, the test's suitability for use in resource-limited or remote settings cannot be adequately evaluated. 

ANSWER: Thank you for this comment. We have added the long-term stability of the reagents in lines 281-282  (page 7).

Comment 8: The manuscript makes strong claims about the test's utility for early detection and outbreak control, but the data presented do not fully support these claims. The limitations in sensitivity, lack of field validation, and insufficient comparison with existing methods undermine the authors' conclusions. How do the authors reconcile the test's limitations with their claims about its potential impact on avian influenza control? The manuscript's conclusions are overly optimistic given the data presented. The authors should temper their claims and provide a more balanced discussion of the test's limitations.

ANSWER: We do not claim early detection in healthy birds with low viral loads, as stated by reviewer 2. In this manuscript, we constantly (see lines 101, 529, 544, 548 and 573) recommend testing fresh dead or sick birds as they have higher viral loads. This sampling and testing protocol follows existing European Food Safety Agency recommendations (see reference 35). However, to minimize the risk of misunderstanding regarding our claim, we have added ‘on samples from fresh dead and sick birds’ and ‘flock-level’ to line 35 (page 1) of the Abstract, and added ‘when using samples of fresh dead or sick birds’ to line 586 of the Conclusion.

We fully agree that when the bird is in its early stage of infection, the viral load might be low for the developed test. For this reason, we already mentioned the importance of testing fresh dead of sick birds (see lines 101, 529, 544, 548 and 573). These birds do not die in first phase of infection and hence act as the red flags for early phase flock infection. To further clarify this for the reader, we have added ‘on samples from fresh dead and sick birds’ and ‘flock-level’ to line 35 (page 1) of the Abstract, and ‘when using samples of fresh dead or sick birds’ to line 586 (page 15) of the Conclusion.

Reviewer 3 Report

Comments and Suggestions for Authors

The reviewed manuscript is dedicated to the design and validation of a novel on-site RT-LAMP-based test for detection of Avian influenza virus subtypes H5, H7, H9. The manuscript itself is well-written and the presented results are promising for implementation of the tests in practice of veterinary laboratories. However, a few questions need to be cleared before the possible publication.

  1. In the current text, it was not stated clearly whether the designed test is specific for low-pathogenic, high pathogenic viral subtypes, or isn’t.
  2. Authors are requested to provide more detailed protocol of the developed assay, because the current explanation seems to be ambiguous in some crucial aspects.
  3. The claimed LoD corresponds Cq values less than 30-32. However, authors mentioned negative results of testing LPAI H5N1 samples with Cq values 29, 30 without further discussion of reasons behind false-negative results.
  4. Authors are requested to compared the designed test with already published analogues and to discuss possible limitations of the assay.

Author Response

Comments and Suggestions for Authors

The reviewed manuscript is dedicated to the design and validation of a novel on-site RT-LAMP-based test for detection of Avian influenza virus subtypes H5, H7, H9. The manuscript itself is well-written and the presented results are promising for implementation of the tests in practice of veterinary laboratories. However, a few questions need to be cleared before the possible publication.

Comment 1: In the current text, it was not stated clearly whether the designed test is specific for low-pathogenic, high pathogenic viral subtypes, or isn’t. 

ANSWER: The Alveo Sense test does not define or differentiate the pathology level of a detected subtype. To clarify this point, we addeddetermination of the cleavage site in relation to its level of pathogenicity’ to line 557-558 (page 15) of the Discussion.

Comment 2: Authors are requested to provide more detailed protocol of the developed assay, because the current explanation seems to be ambiguous in some crucial aspects. 

ANSWER: We added more information about the temperature methodology in lines 134-136 (page 3) and in Figure 2 (page 4).

Comment 3: The claimed LoD corresponds Cq values less than 30-32. However, authors mentioned negative results of testing LPAI H5N1 samples with Cq values 29, 30 without further discussion of reasons behind false-negative results. 

ANSWER: As shown in our results, the LoD is around a Ct of 30-32. However, as our limit of detection is defined as the concentration in which the hemagglutinin target is positive ≥ 95% of the time, detection at or near the LoD range is subject of reaction chemistry and may not exhibit 100% detection: chemistry is sometimes a bit lower (e.g. 29), sometimes a bit higher (33 or 34). By stating a range of LoD, we clearly show that it is not exactly between 30.0 and 32.0.

Comment 4: Authors are requested to compare the designed test with already published analogues and to discuss possible limitations of the assay.

ANSWER: In this paper, we report on the development and laboratory validation of our test, and thus did not perform direct comparisons with antigenic tests or other LAMP-based tests (which are not commercially available anyway). For this reason, we would not like to speculate regarding the performance of other tests, as only a real field experience will determine which tests perform well or not.

Round 2

Reviewer 3 Report

Comments and Suggestions for Authors

Many thanks to authors for their thoughtful comments and careful correction of the manuscript. However, limitations of the applied methodology must be stated for readers not well-accustomed with LAMP to know possible issues when LAMP-based tests are applied. For instance, LAMP needs at least 4 primers, or 6 primers in the full set, and the selection of 4-6 conservative primers in a 200 bp target is challenging, especially, for highly variable pathogens such as influenza viruses. Thus, emerging AIV lineages carrying mutations under primers can become “ghosts” and not be detected by LAMP. The same goes for non-specific amplification and contamination from previous runs. While authors demonstrated that their assay is robust in terms of specificity, one cannot guarantee that these results will be reproducible under real practice conditions on a daily basis, and more extensive trials are necessary to prove absence of false-positive results. Comparison with previously published articles describing AIV-detecting LAMP will also help readers to understand limitations and advantages of LAMP. Without stating sensitivity and specificity of other tests, it is not clear whether clinical parameters of the designed tests are in the normal range for LAMP, or not.

Author Response

Reviewer round 2:

Many thanks to authors for their thoughtful comments and careful correction of the manuscript. However, limitations of the applied methodology must be stated for readers not well-accustomed with LAMP to know possible issues when LAMP-based tests are applied. For instance, LAMP needs at least 4 primers, or 6 primers in the full set, and the selection of 4-6 conservative primers in a 200 bp target is challenging, especially, for highly variable pathogens such as influenza viruses. Thus, emerging AIV lineages carrying mutations under primers can become “ghosts” and not be detected by LAMP. The same goes for non-specific amplification and contamination from previous runs. While authors demonstrated that their assay is robust in terms of specificity, one cannot guarantee that these results will be reproducible under real practice conditions on a daily basis, and more extensive trials are necessary to prove absence of false-positive results. Comparison with previously published articles describing AIV-detecting LAMP will also help readers to understand limitations and advantages of LAMP. Without stating sensitivity and specificity of other tests, it is not clear whether clinical parameters of the designed tests are in the normal range for LAMP, or not.

ANSWERS

Thank you for your thoughts to improve the manuscript.

We have added the following sentence to the discussion to make the readers aware of the complexity of the LAMP PCR design:  “As LAMP uses 4-7 primers for the specific amplification of target DNA, primer design is more complex than for the RT-PCR. This complexity requires careful consideration to ensure that each primer and assay works effectively’ (page 14, lines 507-509.

We have also extended the sentence (page 14, lines 516 to 519) about the importance of consistent monitoring of the reported sequences of the M and H genes: “These findings show that consistent monitoring of the reported sequences of the M and H genes is needed to ensure that the selected primers and probes used in genomic tests as RT-PCR and LAMP assays are capable of detecting all relevant strains’.

In this paper, we report on the development and laboratory validation of our test. We did not make direct comparisons with other LAMP Assays, and thus, would not like to speculate too much about performance comparisons. However, to provide the readers a general impression, we added  the reference Filaire, et al. [18]  to the discussion who reported a comparable LoD of a Ct of around 30 resembling 5.75 to 9.65 viral copies per μL of two 2.3.4.4b HP H5 viruses for a RT-LAMP assay specifically developed for the detection of 2.3.4.4b HPAI H5 strains. (Page 14, lines 526 till 529). As stated in abstract (Page 1, line 36), and discussion (page 15, lines 553-560), only a field validation can show the real diagnostic sensitivity and specificity.